# ITERATIVE MEMORY NETWORK FOR LONG SEQUENTIAL USER BEHAVIOR MODELING IN RECOMMENDER SYSTEMS

## ABSTRACT

Sequential user behavior modeling is a key feature in modern recommender systems, seeking to capture users' interest based on their past activities. There are two usual approaches to sequential modeling : Recurrent Neural Networks (RNNs) and the attention mechanism. As the user behavior sequence gets longer, the usual approaches encounter problems. RNN-based methods incur the problem of fast forgetting, making it difficult to model the user's interests long time ago. The self-attention mechanism and its variations such as the transformer structure have the unfortunate property of a quadratic cost with respect to the input length, which makes it difficult to deal with long inputs. The target attention mechanism, despite having only $O(L)$ memory and time complexity, cannot model intra-sequence dependencies. In this paper, we propose Iterative Memory Network (IMN), an end-to-end differentiable framework for long sequential user behavior modeling. In IMN, the target item acts as a memory trigger, continuously eliciting relevant information from the long sequence to represent the user's memory on the particular target item. In the Iterative Memory Update module, the model early crosses the user behavior embeddings and the item embedding, walks over the long sequence multiple iterations and keeps a memory vector to memorize the content walked over. Within each iteration, the sequence interacts with both the target item and the current memory for both target-sequence relation modeling and intra-sequence relation modeling. The memory is updated after each iteration. The framework incurs only $O(L)$ memory and time complexity while reduces the maximum length of network signal travelling paths to $O(1)$, achieved by the self-attention mechanism with $O(L^2)$ complexity. IMN outperforms various state-of-the-art sequential modeling methods on both public and industrial datasets for long sequential user behavior modeling. It is successfully deployed on an industrial E-commerce recommender system, with a significant 7.29% improvement in the Click-Through Rate (CTR) for a 7-day online A/B test.

## 1 INTRODUCTION

Click-through rate (CTR) prediction is critical for recommender systems. User sequential modeling is the key to mine users' interest for accurate predictions. As the user sequence gets longer, particularly with lengths longer than 1000, the prediction task requires extraordinary long-range dependency modeling, efficient memory storage, acceptable training speed and real-time inference. Recurrent Neural Networks (RNNs) and the long short-term memory (LSTM) are employed by the early sequential recommenders (Hidasi et al., 2016; Hochreiter & Schmidhuber, 1997). Graves et al. (2014) prove that LSTM forgets quickly and fails to generalize to sequences longer than 20. Many empirical results also verify that RNN-based sequential recommenders are surpassed by attention-based methods (Zhou et al., 2017; Kang & McAuley, 2018; Zhou et al., 2018; Pi et al., 2019). Lately, the self-attention mechanism has proven to benefit a wide range of application domains, such as machine translation (Vaswani et al., 2017), speech recognition (Chan et al., 2015), reading comprehension (Cui et al., 2016; Lin et al., 2017) and computer vision (Xu et al., 2015; Parmar et al., 2019). The self-attention mechanism attends to different positions in the sequence, captures the most important features and allows the model to handle long-range dependencies. SASRec adapts the self-

attentive Transformer architecture for sequential recommenders and outperforms convolution-based and recurrence-based methods empirically (Kang & McAuley, 2018).

However, SASRec has a quadratic complexity with respect to the sequence length, limiting its scalability to long sequences. Research on efficient self-attention is based on either sparse attention (Li et al., 2020; Zhou et al., 2020; Child et al., 2019; Kitaev et al., 2020) or approximated attention (Wang et al., 2020), and consequently incompetent against Transformer. SASRec is also unaware of the target item during the encoding process, which could harm its performance.

Deep Interest Network (DIN) is the first architecture that adaptively learns the user interest representation from historical behaviors with respect to a particular target item (Zhou et al., 2018). However, DIN does not model dependencies between elements in the sequence. Research has shown the importance of learning long-range intra-sequence dependencies in sequential modeling (Vaswani et al., 2017). SASRec considers pairwise intra-sequence dependencies and the maximum length of signal traversal paths is $O(1)$, while DIN models no intra-sequence dependencies.

In this paper, we propose the Iterative Memory Network (IMN) for long sequential user behavior modeling. The target item is a memory trigger in IMN, continuously eliciting relevant information from the sequence. IMN early crosses the target item and the user sequence, walks over the sequence for multiple iterations and repeatedly updates the memory vector with new information retrieved from the sequence. The contributions of this paper are summarized as follows:

- We propose the Iterative Memory Network (IMN), an end-to-end differentiable framework for sequential recommenders. To the best of our knowledge, it outperforms the state-of-the-art models for equal sequence lengths, with even more significant advantages on long sequential user behavior modeling. The framework is scalable to sequential recommenders with other objective functions.
- IMN is efficient with $O(L)$ complexity and $O(1)$ number of sequential operations, making it scalable to long sequences. IMN requires significantly less training and inference time, with greater memory efficiency. Implemented on user sequences of length 1000, IMN is deployed successfully on an industrial E-commerce platform with 1500 QPS. The inference time is 30ms. There is a significant 7.29% CTR improvement over the DIN-based industrial baseline.
- To the best of our knowledge, IMN is the first sequential recommender that early crosses the user sequence and the target item. Our ablation study validates empirically that early crossing before further sequence encoding benefits the model performance compared to delayed crossing.
- IMN models both long-range intra-sequence dependencies and target-sequence dependencies in $O(L)$ complexity. In IMN, the memory vector encapsulates the sequence content walked over. The multi-way attention between the sequence, the target item and the memory allows for intra-sequence signal passing. Self-attention achieves this with $O(L^2)$ complexity, and the target attention mechanism allows for no intra-sequence dependency modeling.

## 2 RELATED WORK

**Sequential Recommender Systems**. Sequential recommenders predict the user's next behavior based on his past activities. Recurrent Neural Networks (RNNs) are introduced for early sequential recommenders (Wu et al., 2016; Hidasi et al., 2016). Yet, RNNs are difficult to parallelize and suffer from the problem of fast forgetting (Graves et al., 2014). First introduced in the encoder-decoder framework, attention is shown effective to replace RNNs to encode sequences (Bahdanau et al., 2016; Vaswani et al., 2017). Attention-based recommender systems include self-attention based methods (Kang & McAuley, 2018; Zhou et al., 2017), target-attention based methods (Zhou et al., 2018) and methods based on the integration between RNN and attention (Zhou et al., 2019). Target-attention based methods learn the weights for sequence items with respect to the target item.

**Memory Networks**. Memory Networks have wide applications in Question Answering (QA), finding facts for a query from the knowledge database (Chaudhari et al., 2021). Neural Turing Machines (NTM) introduces the addressing-read-write mechanism for memory searching and update (Graves et al., 2014). Weston et al. (2014) proposes the general architecture for Memory Networks. DMN, DMTN and DMN+ are the subsequent research (Kumar et al., 2016; Xiong et al., 2016; Ramachandran & Sohmshetty, 2017). In recommender systems, MIMN uses GRU as the controller to update user memory slots with each new clicked item (Pi et al., 2019).

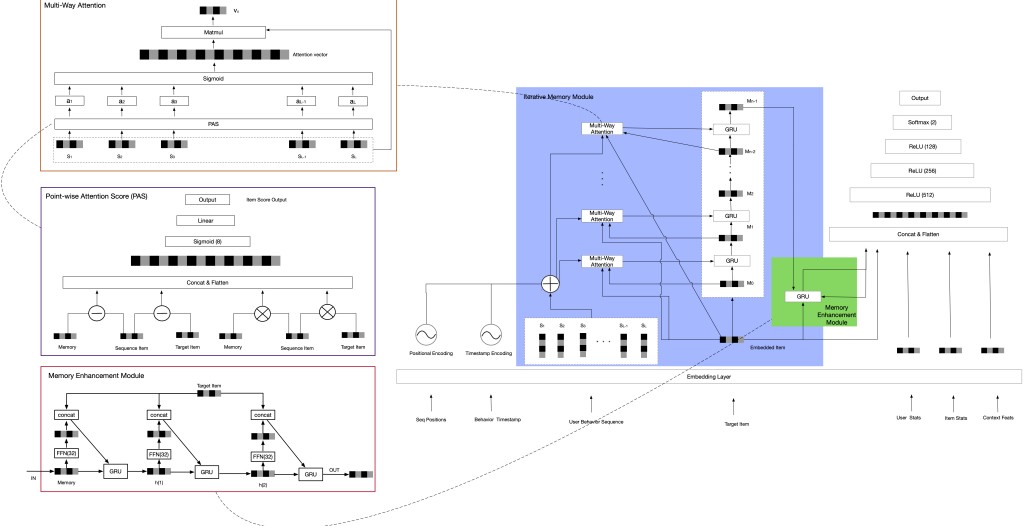

Figure 1: Overview of the Iterative Memory Network (IMN) architecture. The Iterative Memory Update module (in purple) illustrates the repeated memory update process. The details about the multi-way attention and the Memory Enhancement module are on the left.

## 3 PROBLEM FORMULATION

The recommender system models the user-item interaction as a matrix $C = \{c_{mn}\}_{M \times N}$, where $M$ and $N$ are the total number of users and items respectively. The interaction is either explicit ratings (Koren, 2009) or implicit feedback (Agarwal et al., 2009). The CTR prediction task is usually based on implicit feedback. We denote $u \in U$ as user and $i \in I$ as item, and the user $u_m$ clicking on the item $i_n$ makes $c_{mn}$ 1 and others 0. User sequential modeling predicts the probability of a user $u \in U$ clicking on the target item $i \in I$ based on his past behaviors, $i_1, i_2, ..., i_L$, where $L$ is the length of the user sequence. The sequence is usually in chronological order. Our paper focuses on long sequential user behavior modeling, where the length of the user behaviors $L$ is on the scale of thousands. We use the term 'target-sequence dependencies' to refer to dependencies between the target item and the sequence items and 'intra-sequence dependencies' to refer to dependencies between the items within the sequence.

## 4 ITERATIVE MEMORY NETWORK

We illustrate the Iterative Memory Network architecture in Fig.1. The encoder layer converts user behavior features, the temporal features, and the target item features to embedding vectors. The Iterative Memory Update module is the major component in our architecture, modeling both target-sequence dependencies and intra-sequence dependencies. The Memory Enhancement module repeatedly elicits clearer memory with the target item and outputs the final user interest vector. Next, we discuss the framework in detail.

### 4.1 ENCODER LAYER

We use $\boldsymbol{e}_i$ to denote the embedding for the sequence item $i$. Each user behavior consists of not only the clicked item $\boldsymbol{e}_i$, but also the action time. Different users have different action patterns, thus the action time contains important temporal information. Since it is difficult to learn a good embedding directly with continuous time features, we bucketize the time feature into multiple granularities and perform categorical feature look-ups. We slice the elapsed time with respect to the ranking time into intervals whose gap length increases exponentially. In other words, we map the time in range[0,1), [1,2), [2,4), ..., $[2^k, 2^{k+1})$ to categorical features 0,1,2,...$k + 1$. Positional encodings are added to represent the relative positions of sequence items. The timestamp encoding and the positional encoding have the same dimension as that of the item embedding $\boldsymbol{e}_i$ to be directly summed. We

encode the $j_{th}$ user behavior $e_b^j$ as

$$e_b^j = e_i \oplus e_t \oplus e_p \tag{1}$$

where $\oplus$ denotes element-wise sum-up. We denote the embedding for the target item as $v_T$. It shares the item id embedding look-up table with the item id embedding $e_i$ for the sequence.

## 4.2 MULTI-WAY ATTENTION

We calculate the similarities between the sequence item, the target item and the memory. Then we concatenate them into a single feature vector. Similarities are in both Euclidean distance and the Hadamard distance (denoted by $\circ$) to capture multi-faceted similarity measures.

$$\alpha(e, v, m) = [e - m, e - v, e \circ m, e \circ v] \tag{2}$$

where $e$ is the embedding vector for sequence items, $m$ is the memory embedding vector and $v$ is the embedding vector for the target item.

We input the feature vector into a two-layer point-wise feed-forward network, with *sigmoid* as the activation function. Here point-wise feed-forward means the fully-connected feed-forward network is applied to each sequence item separately and identically.

$$a(e, v, m) = \sigma(W^{(2)}\sigma(W^{(1)}\alpha(e, v, m) + b^{(1)}) + b^{(2)}) \tag{3}$$

We have also experimented with the *softmax* activation function for the second layer. There is very slight change in model performance. The multi-way attention mechanism encodes similarities between the user behavior sequence and the target item to extract the user's interest specific to a particular target item. It also encodes similarities between sequence items and the memory. As the memory vector encapsulates the sequence information, the multi-way attention mechanism can model intra-sequence dependencies between items in the sequence and reduce the maximum length of signal traversal paths to $O(1)$.

## 4.3 ITERATIVE MEMORY UPDATE MODULE

In the Iterative Memory Update module, the target item acts as the memory trigger, continuously waking up relevant memory in the sequence. The memory vector is updated after each iteration with the relevant information retrieved from this iteration. We use a GRU to model the memory update process. The initial memory $m^0$ is initialized from the target item vector, to represent that the user's initial memory on the target item is the target item itself.

$$m^0 = v_T \tag{4}$$

For iteration $t$, we calculate the user interest representation $v_u^t$ as a weighted sum pooling of sequence item vectors, with weights derived from Eq.(3).

$$v_u^t = f(v_T, e_b^1, e_b^2, ..., e_b^L) = \sum_{j=1}^{L} a(e_b^j, v_T, m^{t-1})e_b^j = \sum_{j=1}^{L} w_j e_b^j \tag{5}$$

where $e_b^1, e_b^2, ..., e_b^L$ is the list of user behavior embedding vectors, $v_T$ is the embedding vector of target item $T$ and $m$ is the memory embedding vector.

We use the user interest representation $v_u^t$ after iteration $t$ as the input to update the memory $m^{t-1}$.

$$m^t = GRU(v_u^t, m^{t-1}) \tag{6}$$

Since the memory is updated with a weighted sum pooling of sequence item embeddings, the memory contains information on the sequence items. As both the memory and the sequence contains information about the sequence items, the multi-way attention models intra-sequence dependencies. The architecture models co-occurence beyond the $(target\ item, sequence\ item)$ pair. For example, the target item is $rum$ and the user sequence contains $lime$ and $peppermint$. With the target attention mechanism, both the attention weight between the pair $(peppermint, rum)$ and that between the pair $(lime, rum)$ are not high. With our IMN architecture, after the first pass of sequence walk, the memory vector contains information about both the $peppermint$ and $rum$. When it meets the item $lime$ again, the memory of $peppermint$ and $rum$ awakens the item $lime$ since the triplet $(rum, peppermint, lime)$ is the recipe for Mojito and likely to co-occur multiple times in different

samples. Hence, with $lime$ and $peppermint$ in the sequence, the likelihood to click $rum$ increases. While the target attention mechanism finds the items that co-occur frequently with the target item, our IMN finds the composite group of the user's behavior items for the user to click the target item. Furthermore, IMN is structurally different from Memory Networks, which updates the memory with each incoming input and uses the updated memory for computations on the next input. The sequential nature makes it difficult to parallelize, hence infeasible to deploy on recommmender systems that need real-time responses.

## 4.4 MEMORY ENHANCEMENT MODULE

After $N$ iterations of the memory update process, the user memory vector encapsulates both the information in the sequence relevant to the target item $\boldsymbol{v}_T$ and the intra-sequence information. The Memory Enhancement Module enhances the user memory with the target item repeatedly to elicit more clear memory specific to the target item and remove noises.

We use a GRU to model the repeated elicitation process. The GRU's initial state is initialized from the memory after the Iterative Memory Update module, $\boldsymbol{u}_0 = \boldsymbol{m}^N$. For each step, we apply a linear transformation $W^u$ on the GRU's last hidden state, concatenate the transformed vector with the target item, and use the concatenated vector as the GRU's input.

$$\boldsymbol{u}^t = GRU([W^u \boldsymbol{u}^{t-1}, \boldsymbol{v}_T], \boldsymbol{u}^{t-1}) \tag{7}$$

where $\boldsymbol{u}^t$ is the user representation after $t$ steps in the Memory Enhancement Module.

## 4.5 OTHER RELEVANT EXPERIMENTS

*Dot Product Multi-way Attention*. We have experimented with the dot product distance for the multi-way attention. Model performances degrade as expected.

*Sequence Segmentation & Hierarchical Memory Network*. We segment the sequence of length 1000 into sub-sequences of length 100. We compute the user memory vector with IMN for each sub-sequence. The user memory vectors produced are treated as another sequence for IMN. There is no performance improvement.

## 5 EXPERIMENTS

In this section, we present the experimental setups, experimental results, ablation study, model analysis, computational cost analysis, memory efficiency analysis and hyper-parameter choices in detail.

## 5.1 DATASETS AND EXPERIMENTAL SETUP

**Amazon Dataset**. We collect two subsets from the Amazon product data, Books and Movies (McAuley et al., 2015). Books contains 295982 users, 647589 items and 6626872 samples. Movies contains 233282 users, 165851 items and 4829693 samples. We split each dataset into 80% training and 20% test data according to the behavior timestamp. We use the Adam optimizer, with 0.001 learning rate (Kingma & Ba, 2015). The mini-batch size is 512. We use 2 parameter servers and 4 workers, with 10GiB memory for each worker. The MLP layer size is $128 \times 64$.

**Industrial Dataset**. We collect traffic logs from a real-world E-commerce platform. The E-commerce platform has search and recommendation systems, with user click and purchase logs. We use 30-day samples for training and the samples of the following day for testing. There are 1.68 billion training samples and 57 million test samples. MLP layers are $512 \times 256 \times 128$. The hidden state dimensions for GRUs are 32. The mini-batch size is 512. We use the Adam optimizer, with 0.0001 as the learning rate. We use 5 parameter servers and 50 workers, with 75GiB memory for each worker.

**Evaluation Metric**. We use Area Under the Curve (AUC) as the performance measurement metric.

## 5.2 MODEL COMPARISON

- **YouTube DNN**. YouTube DNN uses average pooling to integrate behavior embeddings to fixed-width vectors as the user's interest representation (Covington et al., 2016).

Table 1: Model performance (AUC) on public and industrial datasets

| Model | Amazon Books | | Amazon Movies | | Industrial | |
|---|---|---|---|---|---|---|
| | AUC | Impr | AUC | Impr | AUC | Impr |
| Youtube DNN | 0.8374 | 0.00 | 0.8343 | 0.00 | 0.7353 | 0.00 |
| DIN | 0.8516 | 1.42 | 0.8603 | 2.6 | 0.7375 | 0.22 |
| DIEN | 0.8550 | 1.76 | 0.8654 | 3.22 | 0.7381 | 0.28 |
| SASRec | 0.8214 | -1.60 | 0.8221 | -1.22 | 0.7346 | -0.07 |
| MIMN | 0.8523 | 1.49 | 0.8714 | 3.71 | 0.7377 | 0.24 |
| UBR4CTR | 0.8570 | 1.96 | 0.8796 | 4.53 | 0.7392 | 0.39 |
| IMN 2P | 0.8498 | 1.24 | 0.8821 | 4.78 | 0.7394 | 0.41 |
| IMN 3P | 0.8672 | 2.98 | 0.8835 | 4.92 | 0.7408 | 0.55 |
| IMN 3P+ | 0.8692 | **3.18** | 0.8862 | **5.19** | 0.7423 | **0.7** |

- **DIN**. DIN proposes the target attention mechanism to soft-search user sequential behaviors with respect to the target item (Zhou et al., 2018).

- **DIEN**. DIEN uses attention-based GRU to model user interest evolutions (Zhou et al., 2019).

- **SASRec**. SASRec is a self-attentive model based on Transformer (Kang & McAuley, 2018).

- **MIMN**. MIMN uses fixed number of memory slots to represent user interests. When a new click takes place, it updates the user memory slots with GRU (Pi et al., 2019).

- **UBR4CTR**. UBR4CTR is a two-stage method. The first stage retrieves relevant user behaviors from the sequence with a learnable search method. The second stage feeds retrieved behaviors into a DIN-based deep model (Qin et al., 2020). The Amazon datasets contain no item side information, therefore we use a strengthened version of sequence selection for the first stage with multi-head self-attention on the sequence itself.

- **IMN 2P/3P/3P+**. IMN models without the Memory Enhancement module. 2P refers to 2 iterations of the memory update process, and 3P refers to 3 iterations.

- **IMN 3P+**. 3 memory update iterations, with the Memory Enhancement module. The number of steps $t$ is 3 for the Memory Enhancement module.

## 5.3 EXPERIMENTAL RESULTS

We report model performances on three datasets with maximum affordable sequence lengths in Table 1. We summarize model performances with increasing sequence lengths 50, 100, 200, 500 and 1000 for Amazon Books in Fig.2. Experiments with missing performance data incur Out-of-Memory(OOM) errors that stop training. We have the following important findings:

- *IMN 3P consistently outperforms all state-of-the-art baselines over three datasets.* This demonstrates the effectiveness of our proposed methodology, modeling intra-sequence dependencies and target-sequence dependencies simultaneously. IMN 3P+, with the Memory Enhancement module, has slight improvements over IMN 3P.
- *IMN 3P constantly outperforms SASRec, the Transformer-based sequential recommender, over equal sequence lengths.* As seen in Fig.2, IMN 3P outperforms the compared models over equal sequence lengths. Noticeably, IMN 3P outperforms SASRec over equal sequence lengths affordable by SASRec. This does make sense, considering that SASRec encodes the sequence with multi-head attention with no knowledge on the target item. In contrast, IMN is aware of the target item throughout the encoding process. The ablation study in Section 5.4 verifies the benefits of early cross with the target item.
- *In general, methods that emphatically perform sequential updates seem to have moderate performance gain*. DIEN uses attention-based GRUs to update the user interest with each sequence item. Similarly, MIMN sequentially updates the nearest user memory slots with each sequence item. Fig.2 shows that DIEN constantly outperforms DIN, though the improvement is moderate on sequence length 100.

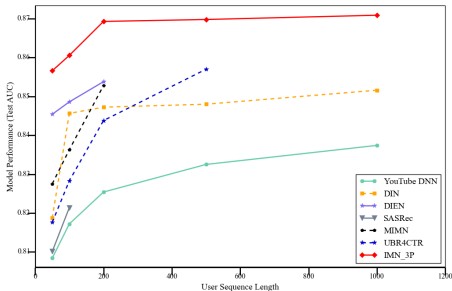

Figure 2: Performance with seq. lengths

| Method | Books AUC | Movies AUC | Industrial AUC |
|---|---|---|---|
| w/o. attention | 0.8374 | 0.8343 | 0.7352 |
| w/o. iterative walk | 0.8516 | 0.8613 | 0.7375 |
| w/o. Euclidean dist. | 0.8549 | 0.8753 | 0.7402 |
| delayed cross w/o. m.e. | 0.8423 | 0.8603 | 0.7389 |
| w/o. m.e. | 0.8672 | 0.8835 | 0.7408 |
| full | **0.8692** | **0.8862** | **0.7423** |

Figure 3: Ablation study on the model structure

## 5.4 ABLATION STUDY

We conduct ablation study about the model structure and report the results in Fig.3. We remove the Memory Enhancement module to produce the ablation model IMN(w/o. m.e.). We further remove the cross with the target item to product the ablation model IMN(delayed cross & w/o. m.e.). We replace the Euclidean distance with another Hadamard distance calculation for the ablation model IMN(w/o. Euclidean dist.). We remove the iterative update process to produce IMN(w/o. iterative walk), which becomes the same as DIN. We replace the attention with average pooling to produce IMN (w/o. attention), which becomes identical to YouTube DNN. The following are our findings:

- *The iterative update process models intra-sequence dependencies, which benefits the model performance significantly.* IMN(full) has a remarkable improvement over IMN(w/o. iterative walk), on par with the improvement of IMN(w/o. iterative walk) over IMN(w/o.attention). It demonstrates the effectiveness of our proposal to model intra-sequence dependencies in addition to target-sequence dependencies as in DIN.
- *Delayed cross with the target item results in performance degradation.* IMN(w/o. m.e.) outperforms IMN(delayed cross & w/o. m.e.) by a large extent, showing that early crossing the user sequence and the target item results in performance gain. This also explains for IMN's performance improvements over SASRec, which crosses the Transformer-encoded user sequence and the target item at the very top.
- *Multi-faceted distance modeling benefits model performances.* IMN(full) outperforms IMN(w.o. Euclidean dist.) to a certain extent, showing that multi-faceted distance modeling is beneficial.

## 5.5 MODEL ANALYSIS

We analyze the compared models and summarize the encoding paradigm, complexity, minimum number of sequential operations and maximum path length in Table 2, with the observations below:

- *IMN is the first architecture with a $(CROSS, ENC)$ structure, which benefits model performances.* We denote the cross between the user sequence and the target item as $CROSS$. We denote sequence encoding as $ENC$. The encoding paradigm for DIN is $(CROSS)$, with no sequence encoding after the target attention cross. DIEN is $(ENC, CROSS)$, encoding sequence items with GRU before interest evolution modeling with respect to the target item. SASRec adapts the Transformer encoder for sequence encoding and is therefore $(ENC, CROSS)$. Similarly, MIMN follows the $(ENC, CROSS)$ paradigm, using a GRU-based controller to update memory slots with the sequence. UBR4CTR is $(CROSS)$ since sequence items cross the target item with no further sequence encoding for the second stage. IMN is the first model with the $(CROSS, ENC)$ paradigm. The user sequence crosses with the target item in Hadamard distance and Euclidean distance before sequence encoding. Ablation study in Fig.3 validates empirically that the early cross benefits model performances.
- *IMN is efficient with $O(L \cdot d)$ complexity and $O(1)$ number of sequential operations, with theoretical shortest maximum path length $O(1)$.* SASRec, the Transformer-based sequential recommender, incurs $O(L^2 \cdot d)$ complexity. IMN incurs $O(L \cdot d)$ complexity, since the optimal number of iterative sequence walk is 3 as seen in Section 5.8. The minimum number of sequential operations measures the amount of parallelizable computations. Pure attention-based methods are at $O(1)$, while recurrence-based methods are at $O(L)$. Maximum path length refers to the maximum length

Table 2: Encoding paradigm, complexity, minimum number of sequential operations, maximum path length for compared methods. $L$ is the sequence length and $d$ is the model dimension.

| Method | Encoding Paradigm | Complexity | Sequential Operations | Max Path Length |
|--------|-------------------|------------|-----------------------|-----------------|
| DIN | $(CROSS)$ | $O(L \cdot d)$ | $O(1)$ | $O(\infty)$ |
| DIEN | $(ENC, CROSS)$ | $O(L \cdot d)$ | $O(L)$ | $O(L)$ |
| SASRec | $(ENC, CROSS)$ | $O(L^2 \cdot d)$ | $O(1)$ | $O(1)$ |
| MIMN | $(ENC, CROSS)$ | $O(L \cdot d)$ | $O(L)$ | $O(L)$ |
| UBR4CTR | $(CROSS)$ | $O(L \cdot d)$ | $O(1)$ | $O(\infty)$ |
| IMN | $(CROSS, ENC)$ | $O(L \cdot d)$ | $O(1)$ | $O(1)$ |

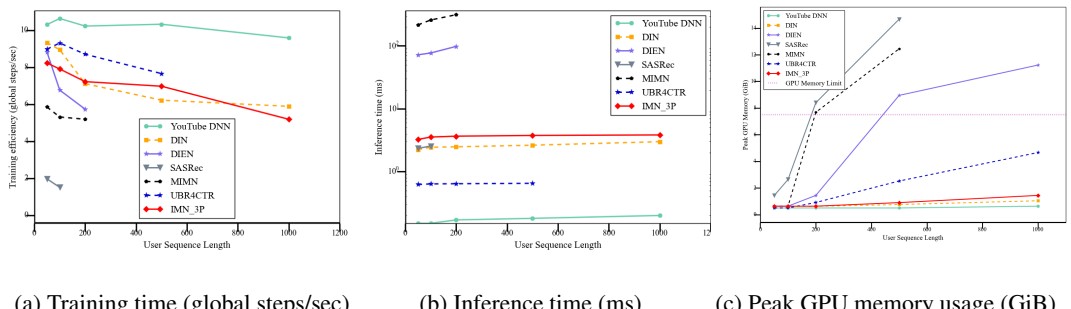

(a) Training time (global steps/sec)  (b) Inference time (ms)  (c) Peak GPU memory usage (GiB)

Figure 4: Computational and memory efficiency analysis

of signal traversal paths. SASRec is $O(1)$ due to the pairwise comparisons in the self-attention mechanism. Our IMN also is $O(1)$ since the memory vector with the sequence information interacts with sequence items through the multi-way attention. There is no intra-sequence signal passing in DIN and UBR4CTR. DIEN and MIMN are at $O(L)$ due to the sequential nature.

## 5.6 COMPUTATIONAL COST ANALYSIS

We report the run-time for compared methods in Fig.4a and the inference time in Fig.4b. The run-time efficiency is measured by global steps per second during training. The real-time efficiency is measured by the inference time in milliseconds. Note that the y-axis of Fig.4b is on the logarithmic scale. For the inference time, we only measure the forward pass cost, excluding the input encoding cost. We use inputs with lengths 50, 100, 200, 500 and 1000. Experiments with missing data incur Out-of-Memory(OOM) errors that stop training. All experiments are conducted on Tesla A100 GPU with 10GiB memory. We summarize our findings below:

- *IMN is computationally efficient with increasing sequence lengths.* IMN involves matrix operations heavily, which are highly optimized to be be parallelizable on GPU. On sequence length 1000, the training time for IMN and DIN is similar, not even doubling that for YouTube DNN. More importantly, the forward pass inference time at sequence length 1000 is only 3.6ms.
- *Methods based on sequential updates are computationally expensive for training and inference.* DIEN and MIMN, the two models based on sequential updates, have significantly lower training and inference speeds. The inference time for MIMN at sequence length 50 is 216ms, while the industrial norm is 30ms to 50ms. In-depth analysis shows that its addressing head is time-consuming. The computational inefficiency forces MIMN to separate the user and the item side, performing user side inference prior to online scoring. At sequence length 100, DIEN's inference time is 77ms. Unlike MIMN, DIEN cannot separate the user and item sides since it performs target attention on top of the GRU encoder. This limits DIEN's scalability to longer sequences.
- *SASRec, the self-attentive method, is efficient during inference time, but not during training time.* The inference speed for SASRec is significantly lower compared to methods with sequential updates. This is expected since self-attention allows for significantly more parallelizations. However, training is relatively slow for SASRec.

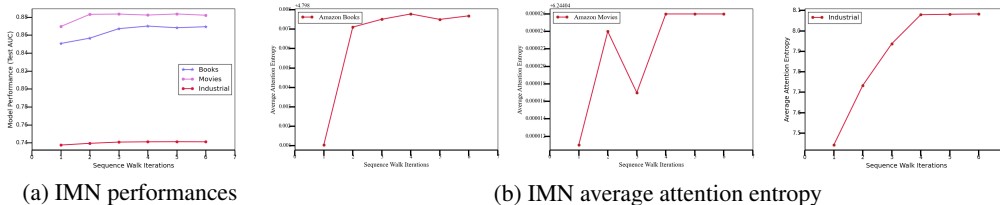

(a) IMN performances        (b) IMN average attention entropy

Figure 5: IMN performances and attention entropy with increasing sequence walk iterations

Table 3: Online A/B result, with Raw denoting the raw data and Impr the relative improvement.

| Method | CTR | | UV-IPV | | TCIC | | CCC | |
|---|---|---|---|---|---|---|---|---|
| | Raw | Impr | Raw | Impr | Raw | Impr | Raw | Impr |
| Industrial baseline | 4.425% | 0.00 | 4.051% | 0.00 | 325741 | 0.00 | 2.979 | 0.00 |
| IMN | 4.748% | 7.29% | 4.373% | 7.95% | 375733 | 15.35% | 3.193 | 7.18% |

## 5.7 MEMORY CONSUMPTION

We evaluate the memory efficiency by measuring the peak GPU memory usage in GiB in Fig.4c. The memory limit is 10GiB. Experiments above the horizontal dotted line in Fig.4c incur Out-of-Memory (OOM) errors. We summarize our findings below:

- *IMN is efficient with memory consumption increasing linearly with sequence lengths.* IMN incurs linear space complexity. The memory overhead is the user memory vector, the same size as the target item. The low peak memory consumption at varying lengths testifies its memory efficiency.
- *Self-attentive methods have the most memory usage increase with increasing sequence lengths. The NTM-based MIMN is also memory-hungry.* SASRec incurs Out-Of-Memory (OOM) errors on sequences longer than 100. Fig.4c also shows that its memory consumption increase is the most substantial with increasing sequence lengths, testifying the $O(L^2)$ memory bottleneck. MIMN is also memory-hungry, since keeping additional user memory slots results in memory overheads.

## 5.8 SENSITIVITY W.R.T NUMBER OF SEQUENCE WALK ITERATIONS

We investigate the impact of sequence walk iterations. Fig.5a shows IMN's performances against the sequence walk iterations. AUCs increase with more iterations and stablize after 3 to 4 iterations. Fig.5b shows the average attention entropy against sequence walk iterations. The trend coincides with that on model performance, indicating that the optimal iteration hyper-parameter is 3.

## 6 ONLINE A/B PERFORMANCE

We report the 7-day A/B test result on the industrial recommender in Table.3. The baseline is a DIN-based deep model with sequences of length 50. IMN is implemented on user click sequences of length 1000. UV-IPV refers to the average click count per user. Total Clicked Items Count (TCIC) is the number of distinct items being clicked. Clicked Categories Count (CCC) is the number of categories clicked per user. TCIC and CCC are diversity measures for recommender systems.

## 7 CONCLUSION

The self-attention mechanism incurs $O(L^2)$ memory complexity that limits the scalability to long sequences. The target attention mechanism, despite having only $O(L)$ complexity, can not model intra-sequence dependencies. Methods with sequential updates incur high computational costs.
IMN models intra-sequence dependencies and target-sequence dependencies simultaneously in $O(L)$ memory and time complexities. It is efficient with computational costs and memory consumption. It outperforms the state-of-the-art sequential recommenders for equal sequence lengths, with even more significant advantages on long user sequential modeling. It is deployed successfully on a real-world E-commerce recommendation platform, with a significant improvement of 7.29% CTR.

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
