# OpenReview forum: "Iterative Memory Network for Long Sequential User Behavior Modeling in Recommender Systems"
_ICLR.cc/2022/Conference — ICLR 2022 Submitted_

### Official Review · Reviewer_NSbP · 2021-10-21

**Correctness:** 3
**Technical Novelty And Significance:** 3
**Empirical Novelty And Significance:** 4
**Recommendation:** 6
**Confidence:** 4

**Main Review:**

Strengths
1. This paper is well written and easy to follow.
2. The topic is significant. Modeling the long sequence is technically challenging.
3. This paper proposes the Iterative Memory Network (IMN), an end-to-end differentiable framework for long sequential user behavior modeling, which shows better performance than baselins. The IMN model is technically sound and interesting.


weaknesses
The compared sequential models lack some STOA, such as [1, 2]. This paper claims its advantages over transformer variants. However, the comparisons ignore the STOA models based on transformers, which does not make sense.


[1] Sun F, Liu J, Wu J, et al. BERT4Rec: Sequential recommendation with bidirectional encoder representations from transformer[C]//Proceedings of the 28th ACM international conference on information and knowledge management. 2019: 1441-1450.
[2] Zhou K, Wang H, Zhao W X, et al. S3-rec: Self-supervised learning for sequential recommendation with mutual information maximization[C]//Proceedings of the 29th ACM International Conference on Information & Knowledge Management. 2020: 1893-1902.


**Summary Of The Paper:**

This paper proposes the Iterative Memory Network (IMN), an end-to-end differentiable framework for long sequential user behavior modeling. Moreover, this paper conducts experiments to show its better performance.

**Summary Of The Review:**

Modeling the long sequence is a hard problem in recommendation applications. This paper proposes the IMN model to learn the user preference from a long sequence and performs better than baselines. The technical significance and novelty of the proposed model are good. The reviewer tends to accept the paper.

---

> ### Author Response · Authors · 2021-11-22
> **Response to Reviewer NSbP on IMN**
>
> We thank the reviewer for finding our topic interesting and our model technically sound. Below we address the reviewer's concerns:
>
> Regarding Transformer-based SOTA models, we have chosen SASRec (Kang et al., 2018) as one of our baselines, which adapts Transformer on sequential recommenders. We have also reviewed Bert4Rec (Sun et al., 2019), but we think this work might not be really impactful over SASRec. Bert4Rec uses the Cloze task (MLM) for next item predictions. In real-time recommender systems, we do not know the items coming after the user's next click item during inference time, leading to a mismatch between training and inference. We perceive Bert4Rec more as a data augmentation technique, yet for large-scale recommender systems, data deficiency is not a major concern. S3-rec (Zhou et al., 2020) validates this point, showing that out of the 6 datasets, SASRec performs better than Bert4Rec in 4 datasets. The suggested paper S3-Rec (Zhou et al., 2020), enhances data representation with self-supervision to learn correlation among the attributes, items and subsequences. It is for the pre-train stage. We recognize it could benefit the model performance. Yet, we think that 1) end-to-end deployable methods should be compared to end-to-end deployable methods 2) the correlations amongst attributes and items differ across datasets, thus the model performance could vary. Nevertheless, we think S3-Rec is a great literature to review to explore directions on pre-train.
>
> Additionally, we have revised the paper entirely. The major revisions include the following:
> 1. Add 7-day online A/B test results on an industrial recommender system with throughput 1500 QPS (Section 6)
> 2. Add computational efficiency analysis, including the training time and the real-time inference time (Section 5.6)
> 3. Add memory consumption analysis (Section 5.7)
> 4. Add ablation study and model analysis (Section 5.4 and Section 5.5)
> 5. Add model performance analysis with increasing sequence lengths (Section 5.3)
> 6. Improve the quality for Figure 1, with detailed visualizations for the Multi-Way Attention and the Memory Enhancement Module.
>
> Thank you again for your review, and for providing relevant references. We would appreciate very much if you could take another look at the revised paper.  Please let us know if you have any additional desired clarifications. Thank you!

---

> ### Author Response · Authors · 2021-11-27
> **Dear Reviewer NSbP - A Gentle Reminder**
>
> Dear Reviewer NSbP,
>
> Thank you very much for the thoughtful review, for finding the model generally sound and for the provision of relevant references. We have thoroughly revised the paper based on inputs from all reviewers. We hope that you would have a chance to read our response to your review as well as the revised paper, before the final discussion phase ends on Nov. 29. We would really appreciate if you would take another look at the revised paper and share additional insights. Please also let us know if there are additional questions, comments, or concerns. Thank you!

---

### Official Review · Reviewer_Q1ux · 2021-11-01

**Correctness:** 3
**Technical Novelty And Significance:** 3
**Empirical Novelty And Significance:** 2
**Recommendation:** 5
**Confidence:** 4

**Main Review:**

This paper proposes an "iterative memory network" for long user sequence modeling such as those for ctr prediction in ads and recommender systems. The basic idea is to use a iteratively updated memory vector to interact with items in the user sequence and target item, so there is higher-order interaction intra-sequence, without the need to use self-attention. Experiments are conducted on two offline datasets comparing with some attention and memory network based methods.

Strength
- Modeling long user sequence is an important problem and is a popular topic recently.
- The proposed model is overall sound and seems to be a reasonable way for this problem.

Weakness
- The major weakness of this paper is evaluation. First, the baselines are relatively out dated. The most recent baseline was in 2019. As mentioned above, the problem is an actively researched area recently, and the DIN paper drew ~400 citations in the last 2 years. The experimental sections need more recent and competitive baselines to compare with. Otherwise the experiments look like what a paper would do 2 years go. Some important related work or baselines seem missing, for example, retrieval based methods such as "User Behavior Retrieval for Click-Through Rate Prediction". Work along this line also try to tackle efficiency issues for long-sequence modeling and need to be discussed and compared with. Second, only offline experiments are done on two datasets. The industrial dataset does not have much details and it is not clear what the impact is. Many papers in this area, such as DIN, performed actual online validation so this paper is on or below the borderline in this dimension.
- The proposed method is intuitive and easy to understand, but may fit an application oriented conference better. Using GRU units to update the memory and establish intra-sequence connections is hand-waving and there is no rigorous mathematical analysis on why and how this works. This may be ok for an application paper but seems to be below the bar of ICLR.

Minor comments
- Figure 1 quality should be better.
- Make notation more consistent, eq4 and eq6 the scripts should be both super or sub.


**Summary Of The Paper:**

This paper proposes an "iterative memory network" for long user sequence modeling such as those for ctr prediction in ads and recommender systems. The basic idea is to use a iteratively updated memory vector to interact with items in the user sequence and target item, so there is higher-order interaction intra-sequence, without the need to use self-attention. Experiments are conducted on two offline datasets comparing with some attention and memory network based methods.



**Summary Of The Review:**

Though the method proposed in this paper is intuitive and sound in general, the lack of rigorous algorithm analysis and significant lack in experiments in terms of setting and baselines may require the paper to be more polished before publication.

---

> ### Author Response · Authors · 2021-11-22
> **Response to Reviewer Q1ux on IMN**
>
> We thank the reviewer for finding the proposal overall sound and reasonable. We have a major revision of the paper based on the review and other reviewers' inputs.
>
> The major paper revisions include the following:
> 1. Add the 7-day online A/B test results on an industrial recommender system with throughput 1500 QPS (Section 6), with additional details on the industrial dataset in Section 5.1.
> 2. Add the computational efficiency analysis, including the training time and the real-time inference time (Section 5.6)
> 3. Add the memory consumption analysis (Section 5.7)
> 4. Add the ablation study and model analysis (Section 5.4 and Section 5.5)
> 5. Add the model performance analysis with increasing sequence lengths (Section 5.3)
> 6. Following your suggestion, we add baseline UBR4CTR, User Behavior Retrieval for Click-Through Rate Prediction (Qin et al., 2020). UBR4CTR is published close to SIM, Search-based User Interest Modeling with Lifelong Sequential Behavior Data for Click-Through Rate Prediction (Qi et al., 2020). Both papers are similar and follow a two-stage paradigm, where during the first stage it selects a subset of sequence items and during the second stage predicts with the shortened sequence. UBR4CTR is academically recognized while SIM is deployed industrially, and we think it is worthwhile to include as our baseline. We intended to add DMAN (Tan et al., 2021) as well. We find that DMAN separates the long sequence and the short sequence, with SASRec-alike self-attentive based mechanisms for long sequences. Since we already have SASRec as our baseline, we decide not to add DMAN.
> 7. Improve the quality for Figure 1, with detailed visualizations of the Multi-Way Attention and the Memory Enhancement Module.
> 8. Make notations for Eq.4 and Eq.6 more consistent.
>
> Below we would like to address the reviewer's concerns:
> 1)  using GRU to update the memory
> - We draw inspirations from the Neural Turing Machines(NTM) in QA tasks. NTM's core architectural component is the controller, which is usually RNN-based and many previous works choose GRU for dynamic context modeling. We have experimented with MLP to substitute GRU, and there is slight decrease in model performances. More importantly, we perceive modeling intra-sequence dependencies as remotely related to using GRU as connections. Rather, it is due to IMN's design to perform multiple iterations of sequence walks. This allows for the pair-wise comparisons between the memory vector and the sequence items.
>
> 2) the possibly better fit with an application oriented conference
> - We have reviewed the research along the same line as IMN, Session-based Recommendations with Recurrent Neural Networks (Hidasi et al., 2016). It applies RNNs on recommenders and has been published in ICLR 2016.
>
> With aforementioned, we perceive our contributions as the following:
>   1. To be best of our knowledge, IMN outperforms the current state-of-the-art sequential models significantly.
>   2. IMN is efficient with the shortest number of sequential operations O(1), theoretical best maximum path length O(1) and the best complexity O(L). This is so far the best as far as our knowledge pertains and not achieved before.
>   3. To be best of our knowledge, IMN is the first architecture following the <CROSS, ENC> paradigm, where the cross between the user sequence and the target item comes before the sequence encoding. Ablation study shows that early crossing before further sequence encoding benefits the model performance (Section 5.4).
>   4. IMN models both long-range intra-sequence dependencies and target-sequence dependencies in linear complexity, reducing to maximum length of signal passing to O(1). This is achieved by the self-attention mechanism with quadratic complexity. Yet the quadratic complexity limits the scalability to long sequences due to the memory bottleneck (Section 5.7).
>   5. From the perspective of applications, we have provided online A/B results. The improvement is very significant in current recommenders. Implemented on user click sequences of length 1000, it is deployed successfully on an industrial recommender system with throughput 1500 QPS. The inference time is 30ms, with a significant 7.29% CTR improvement. Furthermore, we believe the analysis on memory and computational costs are of value to the community.
>
> Thank you again for the detailed and thoughtful review, and for the helpful suggestions. We would appreciate very much if you could take another look at the revised paper. And please let us know if you have any additional comments, questions, or concerns. Thank you!

---

> ### Author Response · Authors · 2021-11-27
> **Dear Reviewer Q1ux - A Gentle Reminder**
>
> Dear Reviewer Q1ux,
>
> Thank you very much for the detailed review, and for the suggestion to refine the evaluation. We appreciate reviewers' valuable comments and have thoroughly revised the paper. Since it is approaching the end of the final discussion phase (Nov. 29), we hope that you would have a chance to read our response to your review as well as the revised paper. We would really appreciate if you would take another look at the revised paper and share additional insights. Please also let us know if there are additional questions, comments, or concerns. Thank you!

---

### Official Review · Reviewer_p4tQ · 2021-11-01

**Correctness:** 4
**Technical Novelty And Significance:** 2
**Empirical Novelty And Significance:** 2
**Recommendation:** 5
**Confidence:** 4

**Main Review:**

Strength

* S1: The writing is clear, easy to follow.
* S2: The authors conducted good experiments, where they also used industrial dataset for comparison. (It will be great if the authors could also provide comparison with current production model(s)).


Weakness:

* W1: The paper has very limited contributions. The idea of memory based attention networks is not new [1, 2].
* W2: The paper has poor analysis (Section 5.4). The paper needs to have deeper analysis in explanation of model architecture and not only performance. For example, besides the benefits of memory based, which were already stated in few papers such as [1, 2], the authors should provide in details about the advantages of the proposed framework (e.g., from theoretical perspectives) compared to previous baselines
* W3: The results in Figure 2 does not seem to be convincing, since the performance on AmazonBooks and AmazonMovies decrease dramatically after only few iterations.


[1] Latent Relational Metric Learning via Memory-based Attention for Collaborative Ranking. WWW 2018.
[2] Signed Distance-based Deep Memory Recommender. WWW 2019.


**Summary Of The Paper:**

This paper focuses on tackle the sequential recommendation task, where the authors proposed Iterative Memory Network (IMN), an end-to- end differentiable framework for long sequential user behaviour modeling. The main contribution of the paper is the IMN framework with efficient in memory and complexity. Specifically, the authors proposed iterative memory updates module with the multi-way attention and memory enhancement module.

**Summary Of The Review:**

The paper is good in terms of writing and conducting experiments. However, the contributions are very limited. This paper requires more in-depth analysis about the model performance, ablation studies, run-time/real-time comparison, etc. A huge drop in the AUC metric after only a few iterations in Figure 2 also raise a concern about the performance.

---

> ### Author Response · Authors · 2021-11-22
> **Response to Reviewer p4tQ on IMN**
>
> We thank the reviewer for finding the writing clear and data-sets well-chosen. Based on reviewers' feedback, we have a major revision of the entire paper and would appreciate very much if you could take another look at the revised paper.
>
> The major paper revisions include the following:
>    1. 7-day online A/B test results on an industrial recommender system with throughput 1500 QPS (Section 6)
>    2. Computational efficiency analysis, including the training time and the real-time inference time (Section 5.6)
>    3. Memory consumption analysis (Section 5.7)
>    4. Ablation study and model analysis (Section 5.4 and Section 5.5)
>    5. Model performance analysis with increasing sequence lengths (Section 5.3)
>    6. For the AUC drop after iterations, we would like to make sure that if it was clear in the paper that the iteration refers to sequence walk iterations rather than training epochs. We previously assigned all IMN models same hyper-parameters, therefore over-fitting occurs with more iterations. We understand the reviewer's concern and have tuned the hyper-parameters by reducing the hidden layer size for MLP for iteration numbers larger than 5 (Figure 5a, page 9).
>    7. Add baseline UBR4CTR, User Behavior Retrieval for Click-Through Rate Prediction (Qin et al., 2020). UBR4CTR is published close to SIM, Search-based User Interest Modeling with Lifelong Sequential Behavior Data for Click-Through Rate Prediction (Qi et al., 2020). Both papers are similar and follow a two-stage paradigm, where the first stage selects a subset of sequence items and the second stage predicts with the shortened sequence. UBR4CTR is academically recognized while SIM is deployed industrially, and we think it is worthwhile to include as our baseline. We intended to add DMAN (Tan et al., 2021) as well. We find that DMAN's contributions lie on the separate modeling of long and short sequences, with SASRec-alike self-attention based mechanisms for long sequences. Since we already have SASRec as our baseline, we decide not to add DMAN.
>   8. Improve the quality for Figure 1, with detailed visualization for the Multi-Way Attention and the Memory Enhancement Module.
>
> We have reviewed carefully the suggested papers.
>   1. Latent Relational Metric Learning via Memory-based Attention for Collaborative Ranking (Tay et al., 2018). The research improves the geometric flexibility of Collaborative Metric Learning (CML) methods with adaptive translation. It is only remotely related to our research. Firstly, it is not on sequential recommender systems, and the user embeddings are not encoded from user sequences. Secondly, the memory vector is to project the user-item relation vector to different latent vector spaces, to increase the geometric flexibility. In contrast, IMN uses the memory vector to abstract user interests from the behavior sequence.
>   2. Signed Distance-based Deep Memory Recommender (Tran et al., 2019). The memory here refers to the embedding process to convert high-dimensional sparse feature vectors to low-dimensional dense feature vectors. Therefore, the memory matrix is the embedding look-up table. This is different from IMN, where the memory vector encapsulates user sequence information related to the target item.
>
> We  think IMN is very different from previous memory-based attentions and perceive our contributions as the following:
>    1. To be best of our knowledge, IMN outperforms the current state-of-the-art sequential models.
>    2. IMN is efficient with O(L) complexity and O(1) number of sequential operations. Implemented on user click sequences of length 1000, it is deployed successfully on an industrial recommender system with throughput 1500 QPS. The inference time is 30ms, with a significant 7.29% CTR improvement.
>   3. To be best of our knowledge, IMN is the first architecture following the <CROSS, ENC> paradigm, where the cross between the user sequence and the target item comes before the sequence encoding. Ablation study shows that early crossing before further sequence encoding benefits the model performance (Section 5.4).
>   4. IMN models both long-range intra-sequence dependencies and target-sequence dependencies in linear complexity, reducing to maximum length of signal passing to O(1). This is achieved by the self-attention mechanism with quadratic complexity. Yet the quadratic complexity limits the scalability to long sequences due to the memory bottleneck (Section 5.7).
>
> Thank you for the detailed review, and for the suggestions to perform more in-depth analysis. We would appreciate very much if you could take another look at the revised paper. And please let us know if you have any additional comments, questions, or concerns.

---

> > ### Comment · Reviewer_p4tQ · 2021-11-28
> > **Thanks for the response**
> >
> > Thanks authors for the response. I have read through the responses, other reviewers' comments, and also the revised version of the paper. It addressed some parts of my concern, so I would like to increase the score of the paper a bit.
> >
> > However, I still have one major concern that is also pointed out by Reviewer UHbH: "In all, I find this paper a good reference for practitioners, especially through the release of source code. But, it does not appear sufficiently novel to my understanding. As other reviewers suggest, this is mostly a multi-layer attention network on top of DIN work, where we also have Wide-and-Deep, Deep-and-Cross, Latent Cross of the same era. They are a bit dated and most commonly published in specialized venues instead of general-audience conferences.".  From my perspective, I do not see the clear motivation of building this architecture, that's the reason I asked for some analyses from theoretical perspectives.
> >
> > Nevertheless, thanks for updating the paper, especially about Section 6: Online A/B Performance.

---

> > > ### Author Response · Authors · 2021-11-28
> > > **Thank you for the further reply! & Further explanation on motivations**
> > >
> > > Thank you very much for the further reply! We would attempt to explain the motivations below:
> > >
> > > ## 1. Perspective from comparison to other SOTA models
> > > We would think of the development of the sequential recommender as the following: 1.Pooling (YouTube DNN, 2016) 2.RNN-based (2016) 3.Attention based, being self-attention based (SASRec, 2018) and target attention based (DIN, 2018) 4.Methods based on sequential updates on top of attention (DIEN, 2019; MIMN, 2019). 5.Other methods build on these core methods with further refinements like time-awareness (TIEN, 2020).
> > >
> > > **Methods based on sequential updates**. Methods based on sequential updates have moderate performance improvements while incur high computational costs. We verify this empirically in Section 5.6.
> > >
> > > **Self-attention based methods**. It refers to SASRec, adapting the Transformer encoder to encode sequences. We think SASRec has the following two major problems.
> > > 1. **The performance is not very promising**. SASRec encodes the sequence with Transformer encoder on the sequence itself, with no knowledge of the target item. The Transformer encoder inputs sequence of length 1000 and outputs sequence of length 1000. The target item is of size 1. To condense the outputs from SASRec, either MLP is used or uni-directional self-attention is used. Neither implementations show promising results on the three datasets. We think **transformer may not be the best encoder for the rank task in recommender system**. We perceive the rank task as a better parallel to Question Answer (QA) task, where the target item resembles the question.  Therefore, our model is designed purposefully for repeated interactions of the sequence and the target. Empirical results testify our modeling advantages.
> > > 2. **Efficiency, especially memory consumption**. In Section 5.7, we have shown the memory bottleneck of self-attention in SASRec. In the SASRec paper itself, it also states "Though our experiments empirically verify the efficiency of our method, ultimately it cannot scale to very long sequences. "
> > >
> > > **Target-attention based methods**. It primarily refers to DIN (2018). DIN does not model for intra-sequence dependencies, and our ablation study shows intra-sequence dependencies modeling benefits the model significantly (Section 5.4).
> > >
> > > ## 2. Perspective from the model structure
> > > As discussed in Section 5.4, IMN is the first model that follows the <CROSS, ENC> structure, crossing the user sequence and the target item at the bottom.  The works brought forward in the review all have their contributions, which are non-overlapping with ours. Wide-and-Deep's contribution is to jointly train the linear model and the deep model. Deep & Cross is the first to demonstrate explicit feature crossing is beneficial. Latent Cross incorporates context embedding into RNN's hidden states for recommenders, where the context refers to the time of day, the location or the user’s device, and not the sequence. IMN is the first structure that crosses the user sequence and the target item, before further encoding. We think our emphasis on repeated interactions between the user sequence and the target item, through the usage of the memory vector, is the first in the recommender system. It is also the first to cross the user sequence and the target item from the bottom before further encodings.
> > >
> > > ## 3. Conclusion
> > > -  From the perspective of comparison to other SOTAs in recommender system, **we propose that IMN is the first sequential recommender that models both intra-sequence dependencies and target-sequence dependencies in O(L) space and time complexity**. It incurs the **shortest number of sequential operations O(1), theoretical best maximum path length O(1) and the best complexity O(L)**. This is so far **the best** as far as our knowledge pertains and **not achieved before**. It allows IMN to accommodate long sequences while other previous SOTAs fail in memory or computational cost.
> > > -  From the perspective of network structure, we propose that IMN is **the first model in user sequential modeling that crosses the user sequence and the target item at the bottom** and explicitly designs for repeated interactions of the user sequence and the target item with the intermediate memory vector. Ablation study in Section 5.4 demonstrates the benefits of early crossing.
> > > -  From the perspective of model performance, IMN **outperforms current state-of-the art sequential models significantly**.
> > > -  From the perspective of applications, we have provided online A/B results. The improvement is **very significant** in current recommenders. Furthermore, we believe **the analysis on memory and computational costs** are of value to the  community.
> > >
> > > With aforementioned, we would like to thank you again for the detailed review, for your appreciation of our efforts, and for the suggestions for more analysis and inclusion of online performance.  They have been of great help for us to improve the paper.

---

> ### Author Response · Authors · 2021-11-27
> **Dear Reviewer p4tQ - A Gentle Reminder**
>
> Dear Reviewer p4tQ,
>
> Thank you very much for the detailed review, and for the suggestion to perform more in-depth analysis. We appreciate reviewers' valuable comments and have thoroughly revised the paper. Since it is approaching the end of the final discussion phase (Nov. 29), we hope that you would have a chance to read our response to your review as well as the revised paper. We would really appreciate if you would take another look at the revised paper and share additional insights. Please also let us know if there are additional questions, comments, or concerns. Thank you!

---

### Official Review · Reviewer_UHbh · 2021-11-06

**Correctness:** 3
**Technical Novelty And Significance:** 2
**Empirical Novelty And Significance:** 3
**Recommendation:** 5
**Confidence:** 4

**Main Review:**

The paper is very clear and the proposals are quite practical. However, I worry that there might not be sufficient academic novelties in this work. Particularly, by giving up the self-attentions within the user's own browsing history, the new method would have fundamental limitations in the comprehension of user intents with long histories. The original self-attention layers would summarize exponentially longer sequences as more layers are added, but the target-attention mechanism only does so linearly. Perhaps this is okay for recommender systems, when there are not much complexity in the sequential patterns, but the contribution is rather technical than academic.

Additionally, the proposed method has another limitation that it is only suitable for reranking tasks when there is a short list of candidates, instead of end-to-end retrieval tasks. This is because every candidate item has to be evaluated through a complex neural network, which is in contrast to traditional transformer models such as SASRec, when the decoder would potentially support the retrieval tasks through approximate max-inner-product search. This follows the same limitation as DIN and I thus worry that its cross-disciplinary impact could be limited.

**Summary Of The Paper:**

The paper talks about a simplification of transformer architectures. The original transformers employ a dense attention between all tokens in a sequence to focus on sequence comprehension. In this work, however, the focus is on the retrieval of the last item. It therefore only evaluates the attention between the proposed candidate item and the preceding items in user history. The other novelty as I see is the replacement of Residual connections between layers with a GRU layer, though the impact is less discussed. The work is successfully deployed in industrial recommender systems, showing significant improvements in the click-through rate.

**Summary Of The Review:**

Interesting practical work and great industrial success, but limited academic contributions.

---

> ### Author Response · Authors · 2021-11-22
> **Response to Reviewer UHbh on IMN**
>
> We thank the reviewer for finding our proposal clear and practical. Below we address the reviewer's concerns:
>
> 1. Our IMN is different from Transformer in terms of the following aspects.
>      1) Network structure. Transformer stacks multiple encoders and feeds the encoded sequence to the decoder. There is no encoder-decoder interaction during the encoder stacking process. Therefore, Transformer has an <ENC,CROSS> structure, where ENC denotes sequence encoding and CROSS denotes the cross between the user sequence and the target item. Our IMN crosses the user sequence and the target item at the very bottom, followed by memory-based attentions. Thus IMN follows a <CROSS,ENC> paradigm. We have done ablation tests to validate empirically that <CROSS,ENC>  is superior to <ENC,CROSS> (Section 5.4, page 6-7).
>     2) Target problem. Transformer is for user intents extraction. Our IMN is for user-item relation modeling, which suits the recommender system scenario better.
>     3) Model performance. Since IMN suits the recommender system better with a relatively superior network structure, it outperforms Transformer for equal sequence lengths (Figure 2, Section 5.3, page 6-7).
>     4) Computational and memory efficiency. IMN incurs O(L) memory and time complexity, making it scalable to long sequences. Transformer's quadratic complexity limits the scalability to long sequences (Section 5.7, page 9)
>
> 2. IMN's contributions are not as limited as perceived to rerank a short list of candidates.
>      1) Rank is the most important task for industrial recommenders. The items for rank is on the scale of 1 to 10 thousands. Empirically, rank contributes to the most Click-Through Rate (CTR) improvement. We have deployed IMN to rank 1500 items in parallel on a real-time industrial recommender system with throughput 1500 QPS and achieved a 7.29% gain in CTR. We have started working on rank size of 10 thousand. Furthermore, retrieval could be done with a mixture of item-to-item based collaborative filtering and deep methods, and the former usually outperforms the latter empirically.
>      2) Using ANN search for retrieval is a compromise for the lack of computational resources on the engineering side rather than algorithmic limitations. The cross between the user and the item benefits model performance empirically, as validated by MLP's superiority over DSSM structures. Our IMN is the first work that showcases early crossing the user sequence and the target item is superior to later crossing. Hence, even if the retrieval task incurs engineering limitations to resort to ANN, deep crossing the user sequence and the target item cannot be escaped.
>
> Additionally, we also have a major revision of the entire paper. The major revisions include the following:
>   1. 7-day online A/B test results on an industrial recommender system with throughput 1500 QPS (Section 6)
>   2. Computational efficiency analysis, including the training time and the real-time inference time (Section 5.6)
>   3. Memory consumption analysis (Section 5.7)
>   4. Ablation study and model analysis (Section 5.4 and Section 5.5)
>   5. Model performance analysis with increasing sequence lengths (Section 5.3)
>   6. Improve the quality for Figure 1, with detailed visualizations for the Multi-Way Attention and the Memory Enhancement Module.
>
> Thank you again for the thoughtful review. We would appreciate very much if you could take another look at the revised paper.  And please let us know if you have any additional comments, questions, or concerns.  Thank you!

---

> > ### Comment · Reviewer_UHbh · 2021-11-27
> > **Concerns not Fully Addressed**
> >
> > First, I thank the authors for their detailed response. However, my concerns are not fully addressed. My concerns are around novelty and fairness of comparisons. Let me reiterate with respect to the author discussions.
> >
> > 1. Novelty. This paper pales in comparison with many related sparse-transformer proposals: dilated-transformer, Fourier-transformer, something-something-transformer, etc. Besides they are earlier, these related papers also target at the frontiers of language modeling for general intelligence - a topic of broader interests. I.e., the results may matter more than the details. Conversely, recommendation is rather a niche topic and the details matter more for a general audience.
> >
> > Also, is 1000 a large number for sequence lengths? I don't know about SASRec, but hugging-face transformer implementation allows for 512 input tokens by default, and these related sparse-transformer works often talk about sequence lengths of 10k+.
> >
> > 2. As I wondered in the initial review, claiming superiority while reducing computation is quite counter-intuitive. Thanks to the authors' response, the reason is pretty clear now. In all the transformer baselines, the user and item interactions are throttled in a bilinear form at the last layer - which the authors denote as the "CROSS" structure. However, in this work, every user-item pair is passed through a deep neural network. This fundamental difference makes the empirical comparisons somewhat misleading, because the "acceleration" at training time is paid back by a much slower inference process, by factors up to 1000 folds depending on the number of candidate targets to evaluate concurrently. The proposed method is like nearest neighbor search, except that the pairwise distances can only be computed through the network on a case-by-case scenario. This is in no ways suitable for retrieval in the original space of millions of items, compared with the baseline encoder/decoder structures. The method might work if limited to a finite space of 1k candidate items, but a fair comparison would be against Transformers with ~1000x latent dimensions, i.e., an embedding size in the hundreds of thousands.
> >
> > I haven't really thought through the training process - does it suffer from a similar slow-down because of the need for pairwise comparisons, instead of automatic gradient aggregation from all candidate items in regular encoder-decoder frameworks? The authors seem to avoid this discussion because they use only explicit-feedback data for training. For broader impacts, the authors may want to discuss the negative sample rates in implicit-feedback scenarios. In those scenarios, a lack of coverage through negative samples may annul the proposed advantages.
> >
> > In all, I find this paper a good reference for practitioners, especially through the release of source code. But, it does not appear sufficiently novel to my understanding. As other reviewers suggest, this is mostly a multi-layer attention network on top of DIN work, where we also have Wide-and-Deep, Deep-and-Cross, Latent Cross of the same era. They are a bit dated and most commonly published in specialized venues instead of general-audience conferences.
> >
> > Sparse-transformer references:
> >
> > [1] Rethinking Attention with Performers. Krzysztof Marcin Choromanski, Valerii Likhosherstov, David Dohan, Xingyou Song, Andreea Gane, Tamas Sarlos, Peter Hawkins, Jared Quincy Davis, Afroz Mohiuddin, Lukasz Kaiser, David Benjamin Belanger, Lucy J Colwell, Adrian Weller. ICLR 2021.
> >
> > [2] Transformer-XL: Attentive Language Models Beyond a Fixed-Length Context. Zihang Dai, Zhilin Yang, Yiming Yang, Jaime Carbonell, Quoc V. Le, Ruslan Salakhutdinov. ACL 2019.
> >
> > [3] Longformer: The Long-Document Transformer. Iz Beltagy, Matthew E. Peters, Arman Cohan. 2020.

---

> > > ### Author Response · Authors · 2021-11-28
> > > **Thank you for the further reply! We hope our further response would at least address some of your concerns.**
> > >
> > > Thank you very much for the further detailed reply. Please allow us to clarify misunderstandings that we see in the discussions.
> > >
> > > 1. We beg to differ on the comparison fairness point from the reviewer. The comparison against Transformer-based SASRec is fair. In fact, IMN is efficient during both training and inference time. Here are the details:
> > >
> > > - We think the reviewer misunderstands the Transformer-based SASRec is exactly the same as the Encoder-Decoder framework and the candidate items **together** constitute the decoder module. Therefore, the reviewer concludes saying that Transformer/SASRec is able to score candidate items in parallel.
> > >
> > >  - With all due respect, this is not true. The recommender system is usually composed of the retrieval stage and the rank stage [1].  You could refer to "System Overview" in [1] for details. We bring this up as a reference to show that the candidate items for the rank stage are not related to each other. The encoder-decoder framework is used in the machine translation task. The encoder could be a sentence in English and the decoder in French. The words in the decoder are related. In contrast, the candidate items to be ranked are not related to each other thus it is not related to the decoder framework.
> > >
> > >  - In fact, there is no existing Transformer-based recommender that works the way that the reviewer thinks, where the candidate items together form the decoder input. We attach SASRec (Self-Attentive Sequential Recommendation, Kang & McAuley, 2018)[2] at the end.
> > >
> > > Therefore, the "automatic gradient aggregation from all candidate items in regular encoder-decoder frameworks" does not exist in current recommender systems. **In fact, the parallel scoring of the candidate items is done in the engine because the candidate items are essentially not related to each other**.  Therefore, **during inference time, the target item for any model is of size 1.** IMN is efficient during inference time, where the forward pass cost is 3.6ms.
> > >
> > > 2. We think the reviewer probably mixes the techniques in the retrieval stage and the rank stage, therefore perceiving that we lack discussions on negative sampling rate, which is a technique for the retrieval stage. The reviewer probably mistakenly thinks that we avoid the discussion on implicit feedback scenarios as we do not discuss negative sampling. This is actually not the case. CTR prediction task is based implicit feedback as seen in Section 3. The negative sampling technique is used for the retrieval stage rather than rank stage. You could refer to [1] for a detailed explanation.
> > >
> > > 2. Thank you for the additional references on efficient Transformers. As we have mentioned, recommender system does not directly adapt the Transformer. We mention specifically the Transformer-based SASrec because we also model intra-sequence dependencies like the Transformer, but only in O(L) complexity. We have also demonstrated the importance of intra-sequence dependency modeling in the ablation study in Section 5.4. We think **since IMN already outperforms Transformer-based recommender in terms of model performance, we do not need to compare against efficient Transformers, whose performance is below the original Transformer**.
> > >
> > > 3. In terms of the novelty and impact, if we have articulated them clearly to the reviewer, we would leave it to the reviewers' discretion. To further ensure that we have conveyed our message clearly, we attach our novelty discussion in the reply "some additional notes on novelty & contributions".
> > >
> > > 4. For the question of whether 1000 is large in terms of sequence length, we would say it is indeed large for the current recommenders, where the norm is the most 50 to 100 recent behaviors. This could be cross-validated in [3] where the sequence length is 100, in [2] where the maximum sequence length is 50 and 200 for two different datasets and in [4] where the sequence length is 50. During our training, sequences of length 250 incurs Out-of-Memory errors for self-attention. The memory bottleneck is a challenge for long sequential modeling, so is computational cost.
> > >
> > > With aforementioned, we would like to thank you again for taking the time with the further detailed review. We hope our reply would address at least some of your concerns.
> > >
> > >
> > > References:
> > > [1] Paul Covington, Jay Adams, Emre Sargin. Deep Neural Networks for YouTube Recommendations. 2016.
> > > [2] Wang-Cheng Kang, Julian McAuley. Self-Attentive Sequential Recommendation. 2018.
> > > [3] Xiang Li, Chao Wang, Bin Tong, Jiwei Tan, Xiaoyi Zeng, Tao Zhuang. Deep Time-Aware Item Evolution Network for Click-Through Rate Prediction, 2020.
> > > [4] Guorui Zhou, Na Mou, Ying Fan, Qi Pi, Weijie Bian, Chang Zhou, Xiaoqiang Zhu and Kun Gai. Deep Interest Evolution Network for Click-Through Rate Prediction, 2019.

---

> > > > ### Author Response · Authors · 2021-11-28
> > > > **Some additional notes on novelty & contributions**
> > > >
> > > > In addition to the reply above, we would like to include some notes on novelty and contributions. For a short summary, you could refer to the **"3.Conclusion"** part of this reply.
> > > >
> > > > ## 1. Perspective from comparison to other SOTA models
> > > > As far as our knowledge pertains, we would think of the development of the sequential recommender with the following: 1.Pooling (YouTube DNN, 2016) 2.RNN-based (2016) 3.Attention based, being self-attention based (SASRec, 2018) and target attention based (DIN, 2018) 4.Methods based on sequential updates on top of attention (DIEN, 2019; MIMN, 2019). 5.Other methods build on these core methods with further refinements like time-awareness (TIEN, 2020).
> > > >
> > > > **Methods based on sequential updates**. They have moderate performance improvements while incur high computational costs. We empirically verify this in Section 5.6.
> > > >
> > > > **Self-attention based methods**. It refers to SASRec, which adapts the Transformer encoder to encode sequences. We think SASRec has the following two major problems.
> > > >
> > > > - **The performance is not very promising**. SASRec encodes the sequence with Transformer encoder on the sequence itself, with no knowledge of the target item. The Transformer encoder inputs sequence of length 1000 and outputs sequence of length 1000. The target item is of size 1. To condense the outputs from SASRec, either MLP is used or uni-directional self-attention is used. Neither implementations on the datasets we experimented on show promising results. Therefore, **we think Transformer may not be the best encoder for the rank task in recommender system**. We perceive the rank task as a better parallel to Question Answer (QA) task, where the target item resembles the question. Therefore, our model places heavy emphasis on the target item, draws inspirations from Neural Turing Machines and is designed purposefully for repeated interactions of the sequence and the target. Empirical results testify our modeling advantages.
> > > > - **Efficiency, especially memory consumption**. In Section 5.7, we have shown the memory bottleneck of self-attention in SASRec. In the SASRec paper itself, it also states "Though our experiments empirically verify the efficiency of our method, ultimately it cannot scale to very long sequences. "
> > > > **Target-attention based methods**. It primarily refers to DIN (2018). DIN does not model for intra-sequence dependencies, and our ablation study shows modeling for intra-sequence dependencies benefits the model significantly (Section 5.4).
> > > >
> > > > ## 2. Perspective from the model structure
> > > > IMN is the first model that follows the <CROSS, ENC> structure, crossing the user sequence and the target item at the bottom. Ablation study in Section 5.4 demonstrates the superiority of early cross. The works mentioned in the review all have their contributions. Wide-and-Deep's contribution is to jointly train the linear model and the deep model for memoization and generalization. Deep & Cross is the first to demonstrate explicit feature crossing is beneficial. Latent Cross incorporates context embedding into RNN's hidden states for sequential recommenders, where the context refers to the time of day, the location or the user’s device, and not the sequence. IMN is the first structure that crosses the user sequence and the target item, before further encoding. We think our emphasis on repeated interactions between the user sequence and the target item, through the memory vector, is the first in the recommender system. It is also the first to cross the user sequence and the target item from the bottom.
> > > >
> > > > ## 3. Conclusion
> > > > -  From the perspective of comparison to other SOTAs in recommender system, **we propose that IMN is the first sequential recommender that models both intra-sequence dependencies and target-sequence dependencies in O(L) space and time complexity**. It incurs the **shortest number of sequential operations O(1), theoretical best maximum path length O(1) and the best complexity O(L)**. This is so far **the best** as far as our knowledge pertains and **not achieved before**. It allows IMN to accommodate long sequences while other previous SOTAs fail in memory or computational cost.
> > > > -  From the perspective of network structure, we propose that IMN is **the first model in user sequential modeling that crosses the user sequence and the target item at the bottom** and **explicitly designs for repeated interactions of the user sequence and the target item with the intermediate memory vector**.
> > > > -  From the perspective of model performance, IMN **outperforms current state-of-the art sequential models significantly**.
> > > > -  From the perspective of applications, we have provided online A/B results. The improvement is **very significant** in current recommenders. Furthermore, we believe **our analysis on memory and computational costs** are of value to the  community.
> > > >
> > > > Thank you for taking the time to read through the long reply. We hope it would partially alleviate your concerns on novelty & impact.

---

> > > > ### Comment · Reviewer_UHbh · 2021-11-29
> > > > **Response**
> > > >
> > > > 1 The authors suggest that the paper focuses on the probability prediction for the binary outcome of every user-item pair, using a heavily pruned sparse attention network to achieve 3.6ms latency per input pair. My point is that this is not quite a sparse transformer, which can also build a user embedding vector that allows fast retrieval from a larger set of items (in the 100ks to millions), in a total time of under 100ms. While 3.6ms is smaller than 100ms in terms of latency claims, as for accuracy claims, we have to also consider that the proposed network uses 36x total computation to rank 1k items for 1 user. To close the gaps, I am not sure if the authors should give Transformers more benefits, such as a larger embedding size, in these accuracy benchmarks.
> > > >
> > > > 2 I am just curious about the actual positive-to-negative sample rate in the paper's experiments. I want to know if the 36x difference affects the claims on training efficiency including negative sampling at up to 1k:1 ratio.
> > > >
> > > > 5 We may quote 512 sequence length (along with some latency claims, but people can do better on GPUs) from this page: https://huggingface.co/blog/bert-cpu-scaling-part-1 Other sparse transformer models may be found on this page: https://huggingface.co/transformers/pretrained_models.html
> > > >
> > > > Overall, the authors may be stuck in the framework of two-stage retrieval systems, but I am really thinking about how they may be combined into a single-stage model. In its current form, the paper shows how to build a better binary predictor in the reranking phase, which yields somewhat narrower impacts.

---

> > > > > ### Author Response · Authors · 2021-11-29
> > > > > **Thank you for the further response!**
> > > > >
> > > > > Thank you for the additional insights.
> > > > >
> > > > > We totally agree with you on the possibility to combine retrieval and rank into a single-stage model. From our perspective, the need for a two-stage paradigm is a compromise for the lack of computational resource on the engineering side for the retrieval task. The retrieval task cannot score items on the scale of billions, therefore it resorts to ANN. It cannot be as accurate as the rank task because it denies the possibility to cross the user and item sides within the model. In fact, if computational resources are sufficient or the number of items are small, we could use one rank task for all.
> > > > > From this perspective, we fully agree on IMN's current impact limited to the rank task. Yet we think its future impact could possibly be larger when the engineering limitations are conquered. Even if not, we have described the importance of the rank task in our first response. Most previous works (DIEN, MIMN, DIN...) are specifically on the rank task as well.
> > > > >
> > > > > Below we elaborate on the detailed points brought up by the reviewer.
> > > > > 1. We concur with you on Transformer's ability to mine user interests.  We propose that IMN outperforms the SOTA models for the rank task, which would be extendable to the large-scale retrieval if engineering limitations are conquered. When we compare against SASRec, the Transformer variant on the recommender system, the comparison is fair. We use the same set of features, same samples, sample experimental setups and the only variable is the model structure. We think comparing the 3.6ms inference time (or multiply by 1k) for IMN against the hypothetical 100ms may not be very informative:
> > > > >    - We think comparing retrieval time and rank time may not be very informative. The retrieval time is not limited to time spent on the forward pass, but also the ANN search time.
> > > > >    - The 1k item size actually is not very relevant to our experiments. When we use either IMN or Transformer/SASRec, the user interest embedding is concatenated with other features like user profile and fed into hidden layers. Therefore, for one single sample, all methods have one target item and thus Transformer/SASRec has no benefits of time saving. During training, the mini-batch size is 512. The 1k-10k statistics are relevant to our online serving.
> > > > >    -  In fact, SASRec does not fall short IMN on inference time (Section 5.6). The major bottleneck for SASRec is the memory (Section 5.7), making it Out-of-Memory (OOM) when sequence length is larger than 200.
> > > > >
> > > > > To conclude on this we think that we cannot compensate Transformer/SASRec's capability on user interest mining with other benefits on the rank task. Nevertheless, we acknowledge this.
> > > > >
> > > > > 2. The positive-to-negative sample rate is 1.326:1. "I want to know if the 36x difference affects the claims on training efficiency including negative sampling at up to 1k:1 ratio." Here we are a little confused. We would like to explain based on our conjectures on the statement. As in our Problem Formulation (Section 3), the positive sample has label '1' and the negative sample has label '0'. There is only 1 target item for each sample, and there is no negative sampling.  The 1k size is only during the inference stage, where we rank 1k items in parallel.
> > > > >
> > > > >
> > > > > 3. "We may quote 512 sequence length (along with some latency claims, but people can do better on GPUs) " Indeed, the performance could vary dependent on the types of GPU (e.g. NVIDIA RTX 2080 Ti  is 37% faster than the 1080 Ti with FP32),  CPU, coding languages and other experimental settings. We quickly read through the reference provided. It also shows PyTorch is more inference-efficient compared to Tensorflow (we use Tensorflow). We could only claim that our experiment is done with "all other things being equal" condition and we have provided the experimental settings in Section 5.1. And if we understand the point correctly, you probably mean that Sparse Transformer might do better in terms of computational efficiency than IMN. We do not deny the possibility, yet we think this does not affect our work: 1)  We think since IMN outperforms Transformer in terms of model performance, we do not need to compare against efficient Transformers that fall short Transformer in model performance.  2) We think adapting the sparse Transformers to recommender system would be another contribution, which parallels SASRec's contribution adapting the original Transformer, and Hidasi's contributribution adapting RNN  (2016). 3)  IMN's complexity is O(L), number of sequential operations is O(1). This is theoretically the best. It is based on analysis, and empirical analysis (Section 5.6 & 5.7) testifies that.
> > > > >
> > > > > Thank you again for the continued interest and the insights shared. We appreciate the references on Sparse Transformers, and the relevant statistics in the references. Please let us know if there are additional desired clarifications.

---

> > > > > > ### Comment · Reviewer_UHbh · 2021-11-30
> > > > > > **Thanks for the discussions**
> > > > > >
> > > > > > I appreciate the detailed replies from the authors. Let me summarize what I learned:
> > > > > >
> > > > > > Pros:
> > > > > > * Explicit cross-features are an easy way to get high accuracy numbers, compared with latent low-rank approaches.
> > > > > > * Within the realm of cross-features, the paper contains some interesting ideas to utilize the sparse / sequential structures of the input features.
> > > > > > * The baseline DIN method in this paper received 600+ citations in the last 4 years. This paper, being better in accuracy, could potentially gather some attentions from practitioners.
> > > > > >
> > > > > > Cons:
> > > > > > * Compared with low-rank approaches, explicit cross-features have fundamental limits in the scalability to large number of target items.
> > > > > > * The 1.3:1 negative rate during training does not reflect well on the true challenges in information-retrieval systems, which are often asked to search for needles in a haystack.
> > > > > > * Sparse transformers are a well-discussed topic in other fields such as NLP. Particularly, the NLP (Sparse-)Transformer models appear to have better memory-efficiency than the SASRec baseline and they should be discussed and/or benchmarked.
> > > > > >
> > > > > > I will keep my scores as is (weak reject), because this conference is more focused on interdisciplinary impacts. I do see values in this work and its associated code release and wish the authors the best of luck in getting them recognized.

---

> > > > > ### Author Response · Authors · 2021-11-30
> > > > > **Thank you for the further response! (Part 2/2 : A short note)**
> > > > >
> > > > > We think we get where the confusion arises. The 1k-10k item number statistics refer to the items being scored during online inference. The current number is 1500. The rank task selects the top 20 from 1500 for the user to view.  The 20 items are items with exposure (a.k.a. users see).Amongst the 20, if the user clicks, it is a positive sample. If not, it is negative. And those not selected from the 1k items are not recorded because they are not presented to the users and therefore no supervision signals.  Therefore, the 1k statistics would not be related to negative samples. Also, amongst the 20 selected, the user could click multiple items, and they are all considered positive samples.

---

> ### Author Response · Authors · 2021-11-27
> **Dear Reviewer UHbh - A Gentle Reminder**
>
> Dear Reviewer UHbh,
>
> Thank you very much for the thoughtful review. We appreciate reviewers' valuable comments and have thoroughly revised the paper.  Since it is approaching the end of the final discussion phase (Nov. 29), we hope that you would have a chance to read our response to your review as well as the revised paper. We would really appreciate if you would take another look at the revised paper and share additional insights. Please also let us know if there are additional questions, comments, or concerns. Thank you!

---

### Decision · Program_Chairs · 2022-01-20

**Decision:**

Reject

**Comment:**

The paper proposes a new architecture named Iterative Memory Network (IMN) to encode long user behavior sequence for recommendations. Reviewers appreciate the clarity of the writing as well as practicality and the O(L) complexity of the proposed architecture, however do raise questions on novelty. Different design choices employed in the paper are not well explained. The rebuttal was not able to convince the reviewers to accept the work at this venue, but reviewers do feel the paper could fly in an application oriented venue.